# Effects of Vocational Rehabilitation on the Use of Health Care Services in Finland: A Propensity Score Analysis

**DOI:** 10.3390/ijerph192315809

**Published:** 2022-11-28

**Authors:** Hanna Rinne, Riku Perhoniemi

**Affiliations:** The Social Insurance Institution of Finland, Nordenskiöldinkatu 12, 00250 Helsinki, Finland

**Keywords:** vocational rehabilitation, occupational rehabilitation, rehabilitation, health care services, outpatient care, register data, propensity score

## Abstract

Vocational rehabilitation may affect the frequency of health care use by improving the access or reducing the need for health care. We examined whether participation in rehabilitation effects the healthcare services use. Register-based data was utilized on all individuals aged 15–60 living in the city of Oulu, Finland, who started vocational rehabilitation in 2014–2015 (N = 784). We examined the use of outpatient health care services from 1.5 years before to 1.5 years after the start of rehabilitation and 1.5 years after the end of rehabilitation, and compared it to the propensity score matched controls. Rehabilitees had on average 1.5 visits to outpatient health care services in the 6th quarter before the start of rehabilitation. In the 4th quarter before the start of rehabilitation, that number increased to 1.8. After the rehabilitation period, the quarterly number of visits returned to the same level as at the beginning of the follow-up. The biggest changes were in the use of occupational health services. Compared to the propensity score matched controls, vocational rehabilitation did not appear to affect the use of health care services. Vocational rehabilitation seems to replace need for other services but not to affect the need to receive treatment for the underlying disease.

## 1. Introduction

The main target of vocational rehabilitation is the promotion of employment [1]. There is indeed some evidence that vocational rehabilitation can have an effect on subsequent work participation [2]. On the other hand, a large proportion of young people who have participated in vocational rehabilitation end up on disability pension or become unemployed [3]. However, rehabilitation may have other outcomes such as improved quality of life and better perceived health [4].

In addition to official goals such as promoting return to work [2], it is valuable to know what wider consequences occupational rehabilitation can have. One of the consequences may be change in the use of health care services. A possible indirect effect from occupational rehabilitation on health care services use can be significant both for individual rehabilitees but also for systems providing supporting health and work ability.

Although a separate service from health care, vocational rehabilitation can affect the use of health services as a by-product of rehabilitation. On one hand, vocational rehabilitation can increase the use of health services in the long run. Work ability problems are often indicative of health problems, and vocational rehabilitation could improve access to health care. This may be due for example to guidance or increased awareness of services. On the other hand, vocational rehabilitation can reduce the use of health services during rehabilitation if it is employed as an alternative to efforts aimed at solving existing work capacity problems through health services. This indirect reducing effect may also appear over the long term if vocational rehabilitation helps to reduce the need for health service use. These possible effects can be viewed as positive if they reflect increased early support for individuals or lower burden on service capacity. To our knowledge, there is no previous research on the relationship between vocational rehabilitation and the use of health services.

To study the effect of rehabilitation, a proper control group for rehabilitees is needed. Because rehabilitees are a highly selected group, it is not useful to compare them to the general population [5]. Due to mandatory nature of vocational rehabilitation, randomized controlled trials are not possible to conduct. Therefore, earlier studies has been used propensity score matching when defining controls for rehabilitees [2,6]. The effect can also be different in different phases of rehabilitation. Using longitudinal individual-level register-based data, the changes in the use of health care services can be studied with the long follow-up period. The changes in level of health care services use before and after the rehabilitation can be compared with difference in differences (DID) approach.

### 1.1. Vocational Rehabilitation in Finland

The aim of vocational rehabilitation is to improve the ability to work or study and to prevent prolonged disability or transition to disability pension. It is targeted to those who have an illness or impairment limiting work or study and who are assessed as likely to benefit from rehabilitation, both to those in and outside of working life.

Vocational rehabilitation is based on an individual plan with a set of defined goals. It consists of various types of rehabilitation such as work and training tryouts, guidance and counselling, education, and training. The duration of rehabilitation may vary from a rehabilitation assessment lasting just a few days to a course of education lasting several years. The most common diagnostic groups causing occupational disability are musculoskeletal and mental disorders, which account for almost 80% of the cases [7,8]. Most of the rehabilitation services are provided free of charge and a rehabilitee may be entitled to a rehabilitation allowance during vocational rehabilitation.

There are two main organizers of vocational rehabilitation in Finland: the earnings-related pension system and the national pensions system administered by the Social Insurance Institution of Finland (Kela). Vocational rehabilitation available through the earnings-related pension system is targeted to those who are integrated into working life. Kela provides vocational rehabilitation for people who have been outside working life for a long time or have just recently or not yet entered the labour market.

Among employees, undergoing vocational rehabilitation has been more common among women, older age groups, and persons with a lower socioeconomic position, as well as those with previous sickness absence [9,10]. Lower occupational status has also predicted lower odds for early rehabilitation [10]. In the working-age population as a whole, secondary school graduates, students and middle-income earners are most likely to participate in rehabilitation organized by the earnings-related pension system [11]. Respectively, young age has predicted participation in vocational rehabilitation through the national pensions system [8].

### 1.2. Factors Affecting the Use of Health Care Services

Finland has a three-sector health care system, where employment status affects the availability of health care [12]. Individuals who are employed have better access to health care services through occupational health care. Unlike other sectors, occupational health care is employer-provided and thus for a major part free of charge to employees. It also provides easy access to outpatient care. In the public sector, offering universal coverage for all residents, patient co-payments are low, but access may be limited and gatekeeping to specialist-level health care strict. Private-sector services are accessible at high co-payment rates.

Additionally, factors related to individual characteristics, for example health and personal barriers such as perceptions of need, attitudes, beliefs and previous experiences with health services, are associated with the use of health care services [13]. Thus, many individual characteristics are found to be associated with the use of outpatient health care services, including income [14], education [15], occupational status [15,16,17,18] and employment status [19]. However, in previous studies the direction of these associations have varied from service to service.

### 1.3. Aims

Vocational rehabilitation is a separate service from health services and is therefore not always seamlessly connected to them. This study examines the transition phase between rehabilitation and health services. To our knowledge, no prior study of that topic exists.

The aims of the study are to examine

How the frequency of health care services use develops from time before the vocational rehabilitation to
time after the start of the rehabilitationtime after the end of the rehabilitationWhether vocational rehabilitation has an effect on the use of health care services

## 2. Materials and Methods

### 2.1. Data

We used a large individual-level register-based dataset of all inhabitants living in Oulu, Finland, in the years 2013–2018 [20]. Oulu is the fifth largest city in Finland, located in the northern part of the country. The inhabitants were obtained from the registers of Statistics Finland. We used StataSE version 14 for the analyses [21].

### 2.2. Intervention Group

Our intervention group included all those who started a vocational rehabilitation period organized by the national pensions or earnings-related pension system between July 2014 and December 2015. We selected the first rehabilitation period and combined successive episodes.

We restricted our analysis to persons 15–60 years old at the end of the year preceding their first spell of vocational rehabilitation. Persons 16 years old are entitled to occupational rehabilitation. The upper age limit was set so that the rehabilitees would not reach the lowest limit of old-age pension in Finland (63 years) during the follow-up. Over 98% of rehabilitees belonged to 15–60 age span. We excluded individuals whose health care use could not be followed for the whole 18-month period after rehabilitation (*n* = 111).

This left us with a dataset that included 784 rehabilitees (see flowchart in Appendix A). The length of the rehabilitation period varied between 2 and 1036 days. The mean length was 300 days (SD 245).

### 2.3. Use of Health Care Services and the Follow-Up Setting

We calculated the number of visit days to a doctor or nurse in outpatient health care services over three-month periods (quarters) for 3 separate 18-month (1.5 years) periods: 6 quarters before rehabilitation, 6 quarters after the start of rehabilitation, and 6 quarters after the end of rehabilitation. Because it was not possible to count reliably the number of visits during the same day, we calculated the number of attendance days for each subject as a proxy for the number of visits.

The outpatient health care services included visits in public primary health care (source of data: registers of the city of Oulu), in occupational health care (registers of the four largest occupational health care providers), in private health care (registers of the Social Insurance Institution of Finland), in public specialized care (Care Register for Health Care, or Hilmo), and emergency visits (registers of the city of Oulu and Hilmo). We examined them separately and in the aggregate.

### 2.4. Background Factors

Individuals’ background factors before vocational rehabilitation were used for matching. These factors were obtained from the registers of Statistics Finland, the Finnish Centre for Pensions and the Social Insurance Institution of Finland. We included sex, age in 5-year age groups, education (tertiary, secondary, primary), main type of activity (employed, unemployed, student, other) and marital status (married, unmarried, divorced/widow) at the end of the previous year, and income quintiles from the previous year. Employment, unemployment and sickness absence days were calculated 0–6, 6–12 and 12–18 months before the start of rehabilitation. Dummy variables (yes/no) for permanent or temporary disability pension and for entitlement to special reimbursement for medicine expenses were counted 0–12 months before the start of rehabilitation. Data on special reimbursements is available for medicines used in the treatment of severe and long-term diseases, indicating a chronic disease [22].

### 2.5. Propensity Score Matching

We used propensity score matching [23,24,25,26] to define controls for the intervention group. For the control group, we randomized the starting day and calculated the ending day of rehabilitation using the same distribution of the length of rehabilitation seen in the intervention group. In matching, we used the above-mentioned background factors and the use of health care services in 6 quarters before rehabilitation. These background variables are presumably associated with the use of health care services. We added most of the variables as categorized and some as continuous. We used nearest neighbour matching 1:1 with a caliper width of 0.015. To examine the balance diagnostic after the propensity score matching, we compared the prevalences and means of background factors between rehabilitees and controls with the standardized difference.

### 2.6. Descriptive Analysis

In descriptive analysis, we examined the average number of visits in outpatient health care services quarterly in each three-month period between (1) 18 months before and after the start of rehabilitation and (2) 18 months before the start of rehabilitation and 18 months after the end of rehabilitation. We analysed rehabilitees separately for outpatient visits in public primary care, occupational health care, private health care, specialized care, emergency, and all outpatient visits combined.

We also compared the use of combined outpatient health care services between rehabilitees and their matched and unmatched controls.

### 2.7. Modelling and Difference in Difference

After matching, we used difference in difference (DID) analysis to measure the differences, between the rehabilitees and their matched controls, in the temporal changes that occur in the use of outpatient health care over time. We modelled the number of outpatient visits with a regression model and compared the average number of visits in quarters 1–2, 1–4 and 1–6 before and after the rehabilitation to study the effect of rehabilitation on the use of health care.

## 3. Results

### 3.1. Distribution of Background Factors

The background factors of rehabilitees, their matched controls and unmatched controls are described in Table 1. The unmatched controls differed systematically from the cases: Rehabilitees were more likely to be women, younger, lower educated and not employed, to have a lower income, and to be unmarried compared to unmatched controls. The rehabilitees had fewer employment days and more unemployment and sickness absence days in the preceding 1.5 years before rehabilitation. In addition, chronic diseases and disability pension were more common for the rehabilitees.

After matching, the distribution of background factors was more similar between rehabilitees and their matched controls. Almost all variables reached the required level of similarity (the standardized difference being <10%). Still, some differences remained. Compared to their matched controls, rehabilitees were more often employed and had more employment days before rehabilitation and more sickness absence days 0–6 months before rehabilitation.

### 3.2. Use of Different Outpatient Health Care Services among Rehabilitees

The quarterly numbers of visits in different health care services before rehabilitation compared to (a) after the start of rehabilitation and (b) after the end of rehabilitation among those who participated in vocational rehabilitation are represented in Figure 1.

The biggest changes took place in the use of occupational health services. The number of occupational health care visits increased steadily before the start of rehabilitation. After the start of rehabilitation (a), the number of visits temporarily decreased. At the end of the rehabilitation (b), the number of visits was first slightly higher but decreased then to the same level as at the beginning of the follow-up. Changes were modest in public health care services. In other services changes were minimal.

### 3.3. Use of Outpatient Health Care Services among Rehabilitees and Their Matched and Unmatched Controls

Figure 2 shows the average total aggregate number of visits during the follow-up. It shows the number of visits starting from before the start of rehabilitation to (a) after the start of rehabilitation and (b) after the end of rehabilitation. In the 6th quarter before the start of rehabilitation, rehabilitees had on average 1.5 visits to outpatient health care services, twice that of the total population. In the 4th quarter before the start of rehabilitation, the number increased to 1.8. In quarters 4–6 before the start of rehabilitation, rehabilitees seemed to have fewer visits compared to their matched controls. Just before the start of rehabilitation in quarters 1–3, the number of visits was at the same level.

After the start of rehabilitation (a), the number of visits among rehabilitees decreased slightly faster than among matched controls, but rose to a higher level than among the controls in quarters 5–6. After the end of rehabilitation (b), rehabilitees had slightly more visits than the controls. The quarterly number of visits returned to the same level as at the beginning of the follow-up.

The use of health services remained clearly higher than in the rest of the population throughout the follow-up period.

### 3.4. Differences in Differences Analysis

Comparing the numbers in the first and second quarters before and after rehabilitation, the average number of visits in outpatient health care services was greater before than after rehabilitation for both rehabilitees and their controls (Table 2). The differences were slightly larger for controls, the DID being still only 0.21 and statistically insignificant (95% CI −0.46–0.88). The comparison was similar as regards the time between the first and fourth quarters. The temporal difference was again slightly larger for controls (DID 0.44, 95% CI −0.77–1.65). When we compared the whole 18-month periods, quarters 1–6, to each other, the difference between before and after vocational rehabilitation was larger among controls than among rehabilitees, but the DID (1.07) was again not statistically significant.

In sensitivity analysis, controlling for the starting year of rehabilitation and the employment days before rehabilitation, which we could not properly match, did not change the DIDs. Only the confidence intervals narrowed slightly.

## 4. Discussion

### 4.1. Main Results and Their Interpretation

The pattern of health care service use changed over the course of vocational rehabilitation. The use of health care services was most common before rehabilitation. The changes were mainly due to visits in occupational health care services. However, compared to the propensity score matched controls, vocational rehabilitation did not appear to have an effect on the use of health care services.

Among rehabilitees, the use of health care services was most common during the year preceding vocational rehabilitation. An injury or illness that requires vocational rehabilitation often also causes a need to use health services. In addition, a doctor’s statement describing a need for rehabilitation because of an illness or impairment is required in order to be admitted into rehabilitation. This requirement may increase the number of visits. The use of health care services dropped to its lowest level during the year after the rehabilitation started, indicating that rehabilitation may replace need for health services.

After the end of the rehabilitation, the use remained at a relatively steady level, the same level as at the beginning of the follow-up. Overall, the number of visits was at a clearly higher level than in the entire population. The reason may be follow-up visits related to rehabilitation, e.g., in occupational health care, or the fact that rehabilitees are unhealthier overall or that they know how to use health services. The purpose of vocational rehabilitation is to improve work ability and the employment outlook, not rehabilitate the illness per se. Thus, the underlying disease causing a need for rehabilitation may remain and require visits to health care even after vocational rehabilitation. In other words, vocational rehabilitation does not necessarily eliminate the illness or impairment that causes work incapacity, but it can reduce its detrimental effects.

Finland’s three-sector health care system also affects the use of health care services over the course of the rehabilitation process. Among the services examined in this study, the biggest changes in the number of visits was seen in occupational health care services. Occupational health care plays a key role in maintaining and promoting employees’ work ability. The most important tasks of occupational health care in relation to rehabilitation are finding out the need for rehabilitation, providing timely guidance for treatment and rehabilitation, and monitoring the ability to cope at work after rehabilitation. The Occupational Health Care Act directs the relevant actors to support work ability at the early stages and to work in close cooperation between the workplace and the occupational health care provider. Unemployed persons and others outside the working life mostly do not have access to occupational health care services, or health services of similar quality and accessibility. Thus, the use of public health services plays a particularly large role among the unemployed [15]. The task of the employment services is to identify unemployed jobseekers who need an assessment of work ability and, if necessary, refer them to the public sector for a health check-up. However, there are shortcomings in the implementation of health check-ups [27] and public healthcare is not always easy to access. Therefore, it is not surprising that the unemployed have an unmet need for health care services [28]. In addition, no one is responsible for coordinating services for the unemployed [27]. This inequity in health service and rehabilitation access may explain why there was no association between vocational rehabilitation and the use of public health services.

Rehabilitees had a smaller decrease in the number of visits than their matched controls when we compared the total number of visits before and after rehabilitation. The smaller decrease among rehabilitees may be due to the fact that rehabilitees have been able to learn to use health care services during the rehabilitation process. This may even be desirable, as unmet need of health services, which is common in certain groups [29], may lead to more severe conditions. The controls could have used other services, for example other rehabilitation such as psychotherapy or rehabilitation organized by the public social and health care system [30], not showing in the study.

The consistent finding of previous studies is that rehabilitees have a higher incidence of illness than the rest of the population, which is natural considering whom the rehabilitation is intended for [9,10]. In Finland, the volume of vocational rehabilitation organized by the national pension scheme is greater than that organized by the earnings-related pension system [7,8]. Probably because of this, the rehabilitees in this study were more often women, younger and lower educated, and more likely to not be employed and to have a lower income, whereas in Sweden, the likelihood for rehabilitation was greater for men and employed persons [5]. As rehabilitation systems differ from country to country [1], comparison between countries is difficult, even between Nordic countries.

### 4.2. Strengths and Limitations

Our study is one of the first to examine occupational rehabilitations’ possible effect on health care use. The main strength of our study was the individual-level register-based data with total population coverage in one city. We were able to cover the whole vocational rehabilitation system, including both the national pension and the earnings-related pension scheme. Previous studies have included only one of these. Our data contained detailed information on the use of health care services. We even had the data for occupational health care services, which is usually not covered by registers. Previous studies on health care services use have mainly used survey data, which is less reliable than register-based data. We also had several background variables for propensity score matching, especially socioeconomic factors and the level of health care use. We managed to find controls for rehabilitees with propensity score matching, with the exception of employment days. However, controlling for employment days in regression models did not affect the results. Because the right to vocational rehabilitation is statutory under the national pensions system, a randomized controlled trial is usually not possible. People cannot be divided into two groups, one of which would receive rehabilitation and the other would not.

A limitation was that the slightly small number of rehabilitees (N = 784), despite that we had all rehabilitees in one city, did not allow any detailed analysis, such as by sex, age group or employment status. However, we were able to consider these in matching. There may also be variation in the need for health care services depending on the diagnosis of rehabilitation. Similarly because of a limited sample size, we had to summarize all different types of vocational rehabilitation, such as work and training tryouts, guidance and counselling, education, or training. However, there may be overlap in the different types of vocational rehabilitation, and it may be random as to which vocational rehabilitation an individual is directed to. In addition, the duration of rehabilitation varied widely, and we could not control for that effect. However, the effect of rehabilitation is not necessarily related to its length. Even short-term rehabilitation can be effective. Further, our data did not include the reasons for outpatient health care visits, which may have differed between the rehabilitees and the controls. Better nationwide registers of health care services use would enable analysis that is more detailed. Because of the data limitations, we excluded individuals whose health care use could not be followed for the whole period after rehabilitation. This might have resulted in an underrepresentation of long rehabilitation periods. We do not assume that it affected the results. It is difficult to compare our findings on the distribution of participants in rehabilitation with previous studies, because they only examined rehabilitation organized by one of the parties, either the national pension or the earnings-related pension scheme [9,10]. Future studies should also include both providers of occupational health care.

## 5. Conclusions

The pattern of health care service use changes over the course of vocational rehabilitation. The changes are mainly due to visits in occupational health care services. The use of health care services is most common around a year before rehabilitation. However, vocational rehabilitation does not appear to have an effect on the use of outpatient health care services. Vocational rehabilitation does not seem to affect the need for treatment of the disease underlying the rehabilitation, at least not yet in this time frame.

Occupational health care seems to be the only service that is connected with the vocational rehabilitation process. For those that do not have access to occupational health care, and rely on public health services, the follow-up in health care services before and after vocational rehabilitation does not seem to be as good.

Since the effects of rehabilitation on health care use may be different in different groups, further studies should examine services separately for different age groups and for those employed and those outside the working life, and should distinguish between different types of rehabilitation and diagnoses. That is possible only with proper register-based data. Policies should focus on providing better health care services for the non-employed and improving co-operation between different services.

## Figures and Tables

**Figure 1 ijerph-19-15809-f001:**
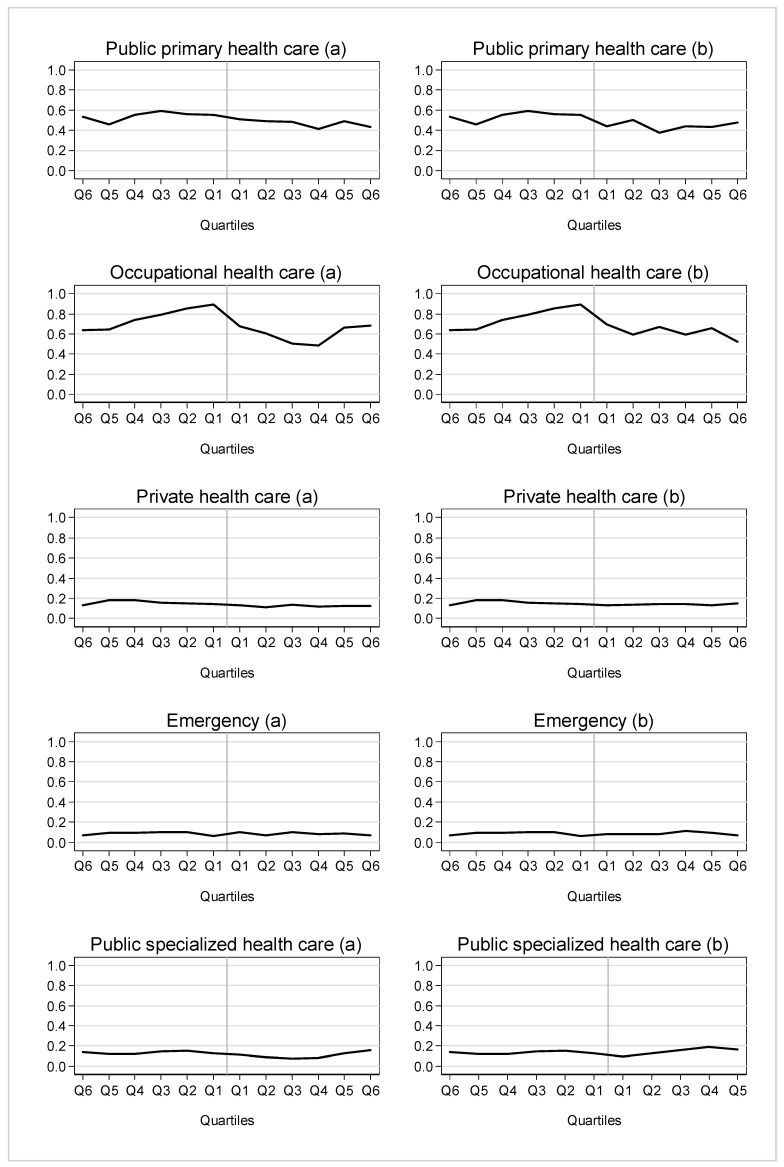
Number of visits in different health services among rehabilitees before the start of rehabilitation and (**a**) after the start of rehabilitation and (**b**) after the end of rehabilitation.

**Figure 2 ijerph-19-15809-f002:**
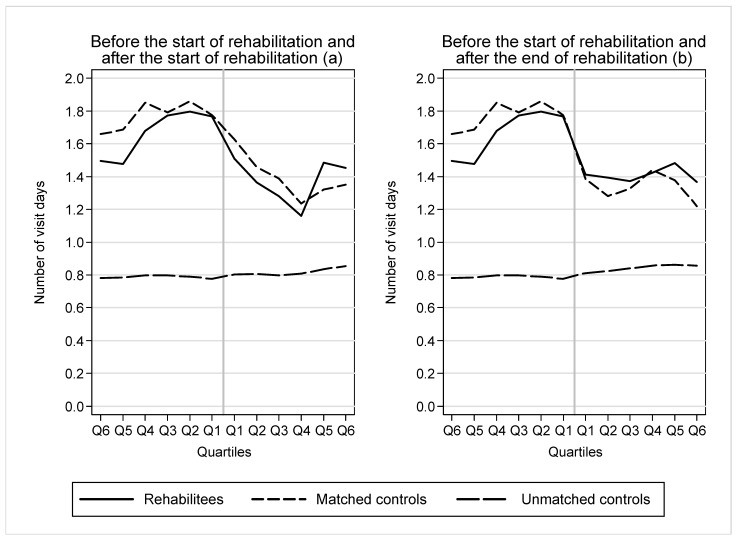
Number of visits in outpatient health care services among rehabilitees and their matched and unmatched controls before the start of rehabilitation and (**a**) after the start of rehabilitation and (**b**) after the end of rehabilitation.

**Table 1 ijerph-19-15809-t001:** Distribution of background factors among rehabilitees, their matched controls and all others.

Variable	Unmatched Controls %/Mean	Matched Controls %/Mean	Rehabilitees %/Mean	Standardized Difference of the Mean
Sex (%)				
Men	52	43	43	−1.0
Women	48	57	57	1.0
Age (%)				
15–19	9	19	16	−9.4
20–24	9	11	11	−1.3
25–29	10	10	11	1.3
30–34	12	8	8	0
35–39	13	7	8	3.8
40–44	11	7	8	1.8
45–49	11	9	8	−0.9
50–54	12	11	13	5.4
55–60	13	18	17	−0.4
Education (%)				
Tertiary	41	21	25	7.7
Secondary	43	43	43	0.3
Primary	16	36	32	−8.8
Main type of activity (%)				
Employed	68	37	42	11.1
Unemployed	13	18	18	−0.7
Student	11	25	22	−8
Other	8	20	18	−6.5
Income quintiles (%)				
1st quintile (lowest)	21	41	37	−9.1
2nd quintile	19	28	26	−5.5
3rd quintile	19	13	14	4.8
4th quintile	20	11	13	5.5
5th quintile (highest)	20	7	9	7.3
Marital status (%)				
Married	43	28	30	4.8
Unmarried	46	59	57	−4.9
Divorced/widow	11	13	13	0.4
Employment days (mean)				
−Q1 & −Q2	123	57	71	16.2
−Q3 & −Q4	121	62	74	14.9
−Q5 & −Q6	122	66	79	15.8
Unemployment days (mean)				
−Q1 & −Q2	23	31	29	−3.9
−Q3 & −Q4	22	27	28	1.4
−Q5 & −Q6	20	27	30	4.1
Sickness absence days (mean)				
−Q1 & −Q2	3	30	37	14.4
−Q3 & −Q4	3	23	27	9.4
−Q5 & −Q6	3	18	20	3.7
Chronic diseases (%)	19	31	31	−0.3
Disability pension (%)	4	19	18	−3.7
Visits in outpatient health care −Q1 (%)				
0	63	44	46	2.9
1	19	19	17	−5.3
2–3	13	19	20	2.4
4–7	4	13	13	−1.4
8+	1	4	4	0.8
Visits in outpatient health care −Q2 (%)				
0	63	45	46	1
1	19	18	18	1.6
2–3	13	19	18	−3.2
4–7	4	13	13	−0.9
8+	1	4	5	1.6
Visits in outpatient health care −Q3 (%)				
0	63	42	42	0.3
1	19	19	19	0
2–3	13	21	21	−0.3
4–7	4	15	14	−3.1
8+	1	3	4	6.2
Visits in outpatient health care −Q4 (%)				
0	63	39	44	9.9
1	19	21	20	−4.2
2–3	13	24	20	−8.9
4–7	4	13	14	0.9
8+	1	3	3	−0.9
Visits in outpatient health care −Q5 (%)				
0	63	45	47	3.6
1	19	20	20	1.3
2–3	13	20	19	−2.1
4–7	4	12	11	−4.4
8+	1	3	3	−2.9
Visits in outpatient health care −Q6 (%)				
0	63	45	47	3.6
1	19	24	23	−0.9
2–3	13	16	17	1.1
4–7	4	11	11	−2.9
8+	1	4	3	−7.7
Total %	100	100	100	
Total N	784	784	88,288	

**Table 2 ijerph-19-15809-t002:** Average number of visits in outpatient health care before the start and after the end of vocational rehabilitation among rehabilitees and their matched controls, the difference between these periods, and the difference in these differences (DID) with 95% confidence intervals.

	Rehabilitees			Matched Controls				
	Before	After	Diff	Before	After	Diff	DID	95% CI
Q1–Q2 before and after	3.57	2.81	0.76	3.64	2.67	0.97	0.21	(−0.46–0.88)
Q1–Q4 before and after	7.01	5.61	1.40	7.28	5.44	1.84	0.44	(−0.77–1.65)
Q1–Q6 before and after	9.98	8.45	1.53	10.62	8.03	2.59	1.07	(−0.67–2.80)

## Data Availability

The data sources of this study were the City of Oulu, the Social Insurance Institution of Finland, the National Institute for Health and Welfare, the Finnish Center for Pensions, four large occupational health care providers, Statistics Finland and the Finnish Tax Administration. The data are not publicly available because legal restrictions apply to the availability of these data, which were used under permissions from third-party data holders for the current study. Permissions to obtain register data from these third-party data holders may be applied for scientific research purposes from the Finnish Health and Social Data Permit Authority Findata (https://www.findata.fi/en/ (accessed on 18 October 2022)). Permission to obtain register data from Statistics Finland may be applied for (https://www.tilastokeskus.fi/meta/tietosuoja/kayttolupa_en.html, (accessed on 18 October 2022)).

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
