# Peer review of "Effects of Vocational Rehabilitation on the Use of Health Care Services in Finland: A Propensity Score Analysis"

_ijerph, 2022, doi:10.3390/ijerph192315809_

Round 1

Reviewer 1 Report

Could you explain more details about vocational rehabilitation? I didn’t understand the meaning as it is written. Does it aim at professionals or patients?

Line 31-41: The authors showed an increase in the use of the health care system, but also a decrease in this use. It's a bit confusing, is positive or negative in each case?

Authors must fuse and arrange the information in one section to avoid repeating information. 

I think the objectives should clarify more. The interest of the study must compare if before the vocational rehabilitation, and after the end (maybe following a time, one month for example) there are differences in the use of the healthcare system, and which differences. 

Authors may include inclusion/exclusion criteria of participants. In addition, a flow diagram of recruitment and study design must include.

Why the age limit is 15 and 60 years old? What are the health problems that cause the inability to work?

What is vocational rehabilitation? What activities do the participants do?

The use of the health care system may be due to the base illness, for example, acute exacerbation in multiple sclerosis. How do the authors control these factors?

There are an intervention group and a control group, why is there no informed consent?

Variables about satisfaction after the end of vocational rehabilitation and the kind of employment must be included. 

The data presentation should be clearer, tables and figures are very unclear. The mean and SD expression of data could be more useful in some variables, such as age. In general, the results are very difficult to understand.  

Reviewer 2 Report

Thank you for giving me the opportunity to review this manuscript. The manuscript is well written, but I would like to suggest a few comments for authors consideration.

The authors do not clarify where they obtain the register of inhabitants from

The authors do not specify the typology of the study, i.e. the study design and its justification.

Reviewer 3 Report

This is an interesting and novel approach to analyze the effect of vocational rehabilitation on outpatient health services use. However, the purpose/aim is not altogether clear in the abstract; methods and analysis results are presented in sufficient detail in the abstract.

I do not find a compelling argument as to why such a study should be undertaken. Is it because no one has conducted this type of research previously? What is compelling about the study of the effect of vocation rehabilitation services onoutpatient use? (Note:authors mention health services used in Lines 31-41). Is propensity score analysis sufficient to justify the study – is it implied based on the title of the paper?

Table 1 is a good start in describing the data. It is helpful if means and standard deviations are presented for the appropriate variables. This complements the results in Table 2. In addition, when percentages are reported it is customary that the number of observations is also reported.

The paper is an example of a carefully considered research design employing matching and a DID analysis.

Section 4.1 articulates the results. The authors also provide commentary (Lines 292 – 303) relative to the difficulty of comparing some of the findings. This section should be placed in Section 4.4.

In Lines 31 - 328 the authors mention the “slightly small number of rehabilitees” and other limitations. First, what is the number? The caveats concerning the remaining study limitations appear to limit the external validity. I would like to see a more nuanced rendering of the limitations justifying the importance of the results.

Round 2

Reviewer 1 Report

the current manuscript is suitable for publication